# Relative burden of lung and pleural cancers from exposure to asbestos: a cross-sectional analysis of occupational mortality in England and Wales

Elizabeth Clare Harris ,[12] Stefania D'Angelo,[12] Andrew Darnton,[3] David Coggon  [12]

¹MRC Lifecourse Epidemiology Unit, University of Southampton, Southampton, UK
²MRC Versus Arthritis Centre for Musculoskeletal Health and Work, University of Southampton, Southampton, UK
³Statistics and Epidemiology Unit, Science Directorate, Health and Safety Executive Bootle Headquarters, Bootle, UK

**Correspondence to**
Dr Elizabeth Clare Harris;
ech@mrc.soton.ac.uk

## ABSTRACT

**Objectives** To explore the extent to which asbestos-exposed jobs vary in the ratio of excess mortality from lung cancer to deaths from pleural cancer.

**Design** Using data on underlying cause of death and last full-time occupation for 3 688 916 deaths among men aged 20–74 years in England and Wales during 1979–2010, we calculated proportional mortality ratios (PMRs), standardised for age and social class, with all occupations combined as reference. For each of 22 asbestos-exposed job groups with significantly elevated PMRs for pleural cancer, we calculated excess mortality from lung cancer (observed minus expected deaths) and its ratio to number of deaths from pleural cancer. To reduce confounding effects of smoking, we adjusted expected deaths from lung cancer in each job group, according to a formula based on its PMR for chronic obstructive pulmonary disease.

**Setting** England and Wales.

**Participants** 3 688 916 men who died aged 20–74 years during 1979–2010.

**Outcome measures** Ratios of excess mortality from lung cancer to deaths from pleural cancer by job group.

**Results** Adjusted PMRs for lung cancer were elevated in all but 4 of the 22 asbestos-exposed job groups, but the ratio of excess lung cancer to deaths from pleural cancer varied widely between job groups, being significantly greater than the overall ratio in six, and significantly less in seven. Analysis for 2001–2010, when (because of changes in coding) ascertainment of pleural tumours was more reliable, showed similar variation between job groups, and indicated an overall ratio of 0.28.

**Conclusions** Excess lung cancer in asbestos-exposed jobs is not in a simple proportion to deaths from pleural cancer, and the ratio may vary importantly according to intensity of exposure to different types of asbestos and concomitant smoking habits. The current burden of lung cancer from occupational exposure to asbestos in Britain may not be so high as previously thought.

## INTRODUCTION

Quantifying the population burden of lung cancer from occupational exposure to asbestos is important for prioritisation of control measures, planning future health-care provision and assessing the impact of

**Strengths and limitations of this study**

► Use of national data covering more than 30 years gave excellent statistical precision.
► Confounding by differences in smoking habits between occupations was addressed by a novel method of adjustment based on proportional mortality ratios (PMRs) for chronic obstructive pulmonary disease (COPD).
► By adjusting PMRs for social class, we reduced the potential for bias because overall mortality in a job group was unusually high or low.
► There was potential for bias from misclassification of occupations and causes of death, but misclassification of lung cancer and COPD as causes of death is likely to have been non-differential with respect to occupation, and therefore to have biased PMRs for those diseases towards the null.
► There was an incomplete ascertainment of pleural cancers before 2001 because deaths ascribed to mesothelioma without any specified anatomical location (most of which would have been pleural) were classed along with other cancers of unknown origin, but a separate analysis for 2001–2010 that included unspecified mesotheliomas supported the main study findings.

preventive strategies. Attributable numbers of deaths have been estimated in several countries including Great Britain,[1 2] Italy,[3] and Argentina, Brazil, Colombia and Mexico.[4] However, the task is complicated by uncertainty about the distribution of exposures across occupations, and the potential for confounding by smoking.

One approach has been to assume that impact is in proportion to the occurrence of mesothelioma. For example, when modelling future numbers of asbestos-related lung tumours in the Netherlands, Van der Bij *et al* applied a multiplier of 1.5 to deaths from mesothelioma[5]—a factor which they derived from an earlier meta-analysis of 55 cohort

studies of asbestos workers.[6] Others have suggested somewhat lower ratios of 0.55,[7] between 0.67 and 1,[1] and 1.1.[3]

One reason for variation in the ratio could be differences in smoking habits, both between countries and within a single country over time, since the combined effects of asbestos and smoking on risk of lung cancer appear to be more than additive.[8 9] In addition, the ratio of excess lung cancer to mesothelioma may vary according to intensity and duration of exposure to different types of asbestos.[6] If so, such variation could lead to differences between asbestos-exposed occupations, according to the nature of their asbestos exposure.

To explore how much the ratio of excess mortality from lung cancer to deaths from mesothelioma differs between occupations, we estimated and compared such ratios for 22 asbestos-exposed job groups, using data from a national analysis of proportionate mortality by occupation in England and Wales. As part of the analysis, we applied a novel method to adjust for potential confounding effects of smoking.

## METHODS

The Office for National Statistics provided us with data on underlying cause of death and last full-time occupation for 3 688 916 deaths among men aged 20–74 years in England and Wales during 1979–2010 (excluding 1981 when records were incomplete). From these, we calculated proportional mortality ratios (PMRs), standardised for age (in 5-year bands), social class (six categories) and calendar period (1979–1990, 1991–2000 and 2001–2010), for occupational categories (job groups) classified as in earlier analyses,[10] taking all occupations combined as the standard.

To address possible confounding by smoking, the prevalence of which varies by occupation, we used PMRs for chronic obstructive pulmonary disease (COPD) to adjust expected numbers of deaths from lung cancer. We first excluded job groups with excess mortality from one or more of COPD, cancer of the pleura or peritoneum, asbestosis or silicosis, which was likely to have arisen from exposures in those jobs (online supplementary table 1). For the 106 job groups that remained (which were presumed to have no major occupational hazard of lung cancer or COPD), we confirmed that the PMR for lung cancer was linearly related to that for COPD by calculation of a Spearman correlation coefficient, and then fitted a weighted linear regression model of the form:

(PMR for lung cancer)=a*(PMR for COPD)+b (1)

For this purpose, the weighting was according to the expected number of deaths from COPD in each job group.

Next, we focused on 22 asbestos-exposed job groups with significantly elevated PMRs over the period 1979–2010 for cancer of the pleura (ICD9 163, ICD10 C38.4, C38.8 and C45.0, lower 95% confidence limit >100) (online supplementary table 2). For these job groups, we used the regression coefficients, a and b, from equation

(1) to adjust expected numbers of deaths from lung cancer according to the PMR for COPD. Thus, the expected number of deaths was multiplied by {a*(PMR for COPD)+b}.

With this correction, we calculated the excess of lung cancer for each job group (observed–expected deaths), and its ratio to the observed number of deaths from cancer of the pleura. CIs for ratios were computed through random simulations (1000 per estimate) in which we assumed that the expected number of deaths from lung cancer was constant, while a number of deaths from lung cancer and cancer of the pleura each followed a Poisson distribution with mean equal to the observed number of deaths from that cancer in our dataset.

During 1979–2000, when ICD9 was used to classify causes of death, there was no separate diagnostic category for mesotheliomas with unspecified anatomical origin, and they were included in a much larger grouping of 'malignant neoplasms without specification of site'. However, ICD10, which was used during 2001–2010, included unique codes for mesothelioma including C45.9 for 'mesothelioma unspecified'. In a sensitivity analysis, we repeated our calculations for this period, aggregating all deaths from mesotheliomas other than of the peritoneum (C45.2, C45.7 and C45.9) with those from pleural cancer.

In addition, PMRs for deaths where mesothelioma was mentioned anywhere in the death certificate text were available for the periods 1980, 1982–2000 and 2002–2010, from national statistics published by the Health and Safety Executive.[11] In further sensitivity analyses, we related excess mortality from lung cancer by job group to excess deaths from mesothelioma in these data (adjusting the ratios to account for there being slightly fewer years of data on mesothelioma).

## PATIENT AND PUBLIC INVOLVEMENT

This research was done without patient or public involvement.

## RESULTS

In the 106 job groups with no major hazard of COPD, silicosis or asbestos-related disease, PMRs for lung cancer correlated strongly with those for COPD (Spearman correlation coefficient=0.78, figure 1). The weighted regression equation was:

(PMR for lung cancer)=0.57*(PMR for COPD)+42.

When the coefficients from this equation were used to adjust expected numbers of lung cancer deaths in the 22 job groups with significantly high PMRs for pleural cancer (online supplementary table 2), the PMR for lung cancer was elevated in all but four, and the overall excess of lung cancer was 1.69 times the number of deaths from pleural cancer. However, the ratio between excess deaths from lung cancer and deaths from pleural cancer varied between job groups, such that in six it was significantly

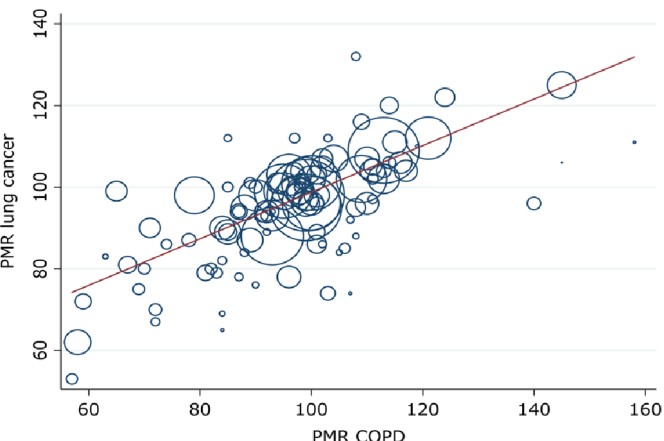

**Figure 1** PMRs for lung cancer and COPD in job groups with no major occupational exposure to causes of either disease: men in England and Wales aged 20–74 years, 1979–1980 and 1982–2010. The areas of the circles represent the expected number of deaths from COPD in each job group over the study period. The regression line of PMR for lung cancer against PMR for COPD is from an analysis that weighted according to the expected number of deaths from COPD in each job group over the study period (see text). COPD, chronic obstructive pulmonary disease; PMR, proportional mortality ratio.

greater than the overall average and in seven significantly less (figure 2). For completeness, online supplementary table 3 shows this job-specific ratio stratified also by time period (1979–1990, 1991–2000 and 2001–2010).

During 2001–2010, 3061 deaths from mesotheliomas other than of the peritoneum were recorded in the 22 asbestos-exposed job groups of interest, in addition to the 1205 classed as pleural cancer. Inclusion of these additional deaths in our calculations gave a lower overall ratio (0.28), but again indicated substantial heterogeneity between job groups (figure 3). Moreover the job groups with the highest and the lowest ratios were much the same as in the previous analysis.

Similar results were obtained in the analysis based on deaths with any mention of mesothelioma in the death certificate text. The overall ratio (in this case to excess rather than total deaths from mesothelioma) was 1.13 for the full study period, and 0.46 for 2001–2010, with similar variation in the ratios for specific job groups.

## DISCUSSION
Our analysis indicates that among occupations entailing exposure to asbestos, the ratio between excess deaths from lung cancer and deaths from pleural cancer/mesothelioma can vary substantially. This suggests that burdens of lung cancer attributable to asbestos are not in a simple proportion to numbers of mesotheliomas, and that the ratio may vary importantly according to the pattern of exposures within a population, and perhaps also smoking habits.

We limited our investigation to men since asbestos-related disease was much less frequent among women.

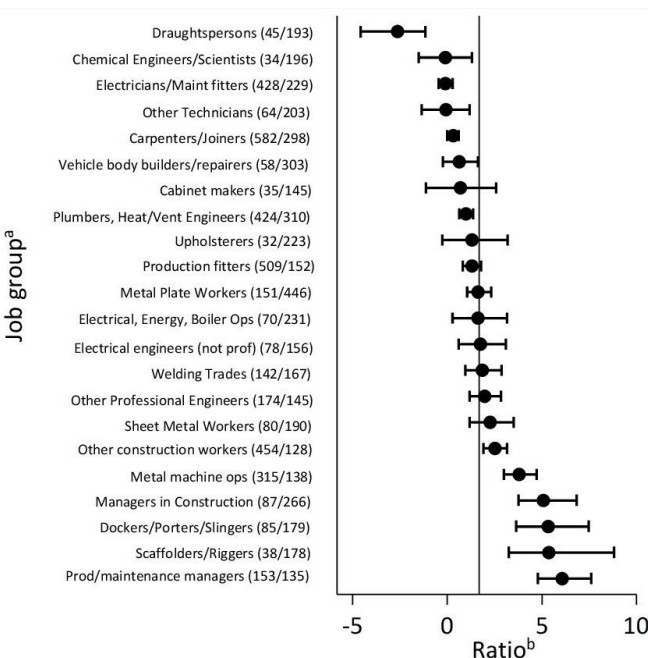

**Figure 2** Ratios of estimated excess deaths from lung cancer to observed deaths from cancer of pleura, 1979–1980 and 1982–2010. [a]Figures in brackets are observed numbers of deaths/corresponding PMRs for cancer of the pleura. [b]Bars represent 95% CIs, and the vertical line indicates the average ratio across all 22 job groups of 1.69.

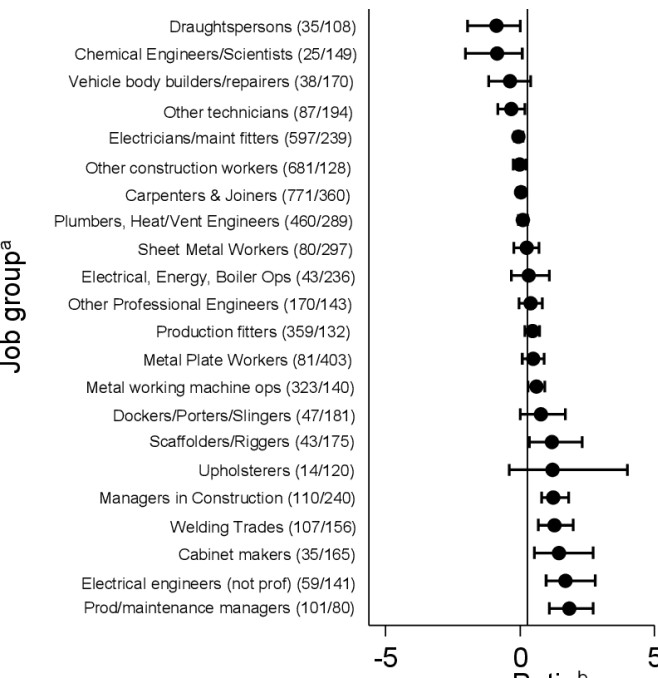

**Figure 3** Ratios of estimated excess deaths from lung cancer to observed deaths from cancer of pleura and mesothelioma, 2001–2010. [a]Figures in brackets are observed numbers of deaths/corresponding PMRs for cancer of the pleura and mesothelioma. [b]Bars represent 95% CIs, and the vertical line indicates the average ratio across all 22 job groups of 0.28.

Moreover, only 30% of the women who died in the early part of the study period (1979–1990) had occupations recorded on their death certificates.[12]

Our use of national data covering more than 30 years gave excellent statistical precision, but there was potential for bias from misclassification of occupations and causes of death. In the UK, death certificates document only the last full-time occupation, but for chronic diseases with long induction periods (such as lung and pleural cancer), jobs held earlier in life may be more relevant. Furthermore, occupations and causes of death are not always assigned accurately.[13] Nevertheless, we think it unlikely that such errors could account for the variation in ratios of excess lung cancer to pleural cancer that we observed.

The 22 job groups on which we focused in our main analysis were those that we could be reasonably confident were associated with an asbestos hazard. However, it was not essential that they should account for all asbestos-related cancer in the study population. Any underascertainment of cases attributable to work in those jobs, either because of migration to other employment or through misclassification of occupations on death certificates, would reduce both the excess mortality from lung cancer and the number of deaths from pleural cancer. However, it would not be expected to bias the ratio of those measures differentially across job groups.

Misclassification of lung cancer and COPD as causes of death is likely to have been non-differential with respect to occupation, and therefore to have biased PMRs for those diseases towards the null. It is reassuring, however, that after exclusion of job groups with exposure to known causes of lung cancer and/or COPD, we observed a strong correlation between PMRs for the two diseases (r=0.78). This suggests that such misclassification was not a major problem.

A greater concern was the incomplete ascertainment of mesotheliomas before 2001 in our main dataset. This occurred because at that time, deaths ascribed to mesothelioma without any specified anatomical location (most of which would have been pleural) were classed along with other cancers of unknown origin. Data from 2001 to 2010, when they were assigned to a specific code, indicated that they outnumbered deaths ascribed to pleural cancer more than twofold. Thus, variation in the extent of underascertainment by job group could have caused serious bias. However, when we restricted our analysis to 2001–2010, and included mesotheliomas other than of the peritoneum with pleural cancers, there was still marked variation in their frequency relative to excess lung cancer. And importantly, the job groups with the highest and the lowest ratios were much the same. Moreover, similar heterogeneity was observed in our analysis based on deaths with any mention of mesothelioma on the death certificate.

As with all analyses of proportionate mortality, there was a possibility that expected numbers of deaths from specific causes of death could be biased if overall mortality in a job group were unusually high or low. However, in stratifying our analyses by social class, we reduced the potential for large variation between job groups in total mortality, and it seems unlikely that such bias could explain differences in the ratio of excess lung cancer to pleural cancer of the magnitude that we observed.

A particular challenge in studying occupational mortality from lung cancer is the scope for confounding by differences in smoking habits between occupations. To address that problem, we adjusted expected deaths from lung cancer according to the PMR for COPD in the job group under consideration. In deriving the formula for the adjustment, we took care to exclude job groups with exposure to major occupational causes of either COPD or lung cancer, in the expectation that the variation between job groups in PMRs would then be driven largely by differences in smoking. The strength of the correlation that we found between the two diseases supported that assumption, and although not all cases of COPD are picked up from death certificates (because of competing causes of death), it seems that the PMR from COPD did provide a meaningful proxy for smoking, making our expected numbers of deaths more reliable than would have been the case without adjustment.

We know from other research that smoking and asbestos interact in causing lung cancer, such that relative risks from the two causes in combination are more than additive.[8 9] It follows that in a person with lung cancer who has been both a smoker and exposed to asbestos, the disease may be attributable to both causes (or put another way, avoidance of either of the exposures might have been sufficient to prevent the disease). However, with the method of statistical analysis that we employed, interactions between smoking and asbestos could be ignored. The parameter on which we focused was the difference between the number of deaths from lung cancer that actually occurred in the job group and the number that would have been expected if the job group had the smoking habits that it did, but no exposure to asbestos. That measure will have included excess deaths attributable to asbestos alone in non-smokers, and to the joint effects of smoking and asbestos as compared with smoking alone in smokers.

The variability that we found in the ratio of excess lung cancer to mesothelioma by job group may in part reflect differences by type of asbestos. Previous meta-analysis of cohort studies has suggested a lower ratio for crocidolite (0.7) than for chrysotile (6.1), amosite (4.0) and mixed fibres (1.9).[6] However, intensity and timing of exposure could also be a factor, and might explain why, when we included mesotheliomas other than of the peritoneum, the mean ratio that we observed across all 22 job groups (0.28) was relatively low. Another analysis, based on national data for England and Wales during 1980–2000, suggested an intermediate ratio in the order of 0.67–1.0.[1] The disparity from our estimate may in part reflect differences in the methods used to control for confounding effects of smoking, but there may also have been changes over time in patterns of

exposure to asbestos, and a reduction in the prevalence of smoking in asbestos-exposed occupations (if the joint effect of asbestos and smoking on lung cancer is more than additive, then a given exposure to asbestos will cause more lung cancers in smokers than in the same number of non-smokers). The lower ratio that we observed suggests that the current burden of lung cancer from occupational exposure to asbestos in Britain may not be so high as previously has been thought.[2]

The potential for variability in the ratio of excess lung cancer to mesothelioma should be taken into account when estimating population burdens of the disease from occupational exposure to asbestos.

**Acknowledgements** We thank the staff of the Office for National Statistics, who provided us with the data files for our analysis, and Vanessa Cox, who helped with computer analysis.

**Contributors** DC designed the study, ECH and DC acquired the Office for National Statistics data, AD acquired the Health and Safety Executive data, and all authors developed the methodology and analysis. SDA and AD carried out the analyses, ECH and DC wrote the first draft of the manuscript, and all authors revised and approved the final version.

**Funding** This work was supported by Unit core funding from the Medical Research Council grant numbers 10.13039/5011000002 65, MRC_MC_UP_A620_1018, MRC_MC_UU_12011/5 (https://mrc.ukri.org/).

**Competing interests** None declared.

**Patient and public involvement** Patients and/or the public were not involved in the design, or conduct, or reporting, or dissemination plans of this research.

**Patient consent for publication** Not required.

**Provenance and peer review** Not commissioned; externally peer reviewed.

**Data availability statement** The data underlying the results presented in the study can be made available to other researchers subject to agreement from the Office for National Statistics. Data relevant to the specific occupations in this analysis are provided in the supporting information file: Supplementary Table 2. 2001-2010 occupational mortality data for England and Wales are available at https://www.ons.gov.uk/peoplepopulationandcommunitypeoplepopulationandcommunity/healthandsocialcare/healthinequalities/adhocs/007958occupationalmortalityinenglandandwales2001to2010007958occupationalmortalityinenglandandwales2001to2010 1991-2000 data available at https://webarchive.nationalarchives.gov.uk/20160129235354/http://www.ons.gov.uk/ons/publications/re-reference-tables.html?edition=tcm%3A77-168405

**ORCID iDs**
Elizabeth Clare Harris http://orcid.org/0000-0001-8037-566X
David Coggon http://orcid.org/0000-0003-1930-3987

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
