## [Reviewer comments · BMJ Open]

ARTICLE DETAILS

TITLE (PROVISIONAL)	Relative burden of lung and pleural cancers from exposure to asbestos: a cross-sectional analysis of occupational mortality in England and Wales
AUTHORS	Harris, Elizabeth; D'Angelo, Stefania; Darnton, Andrew; Coggon, David

VERSION 1 – REVIEW

REVIEWER	Dario Mirabelli Formerly: Cancer Epidemiology, CPO-Piemonte and University of Turin. Now retired I served as expert witness for the public prosecution office in several criminal trials on the death from mesothelioma or lung cancer of asbestos exposed workers
REVIEW RETURNED	26-Dec-2019

GENERAL COMMENTS	In this paper an original approach has been applied to estimate the lung cancer burden attributable to occupational exposures to asbestos, based on England and Wales mortality data, 1979-2010: (i) occupation-specific estimates have been computed for 22 asbestos-exposed occupational groups, where (ii) the expectation for lung cancer had been indirectly adjusted for group-specific smoking prevalence by accounting for COPD mortality. I have a few, minor remarks. In the methods the following definitions have been given: “mesotheliomas other than of the peritoneum (C45.2, C45.7 and C45.9)” (page 6, 54) and “cancer of the pleura (ICD9 163, ICD10 C38.4, C38.8 and C45.0)” (page 6, 12). I understood that pleural mesotheliomas (C45.0) had been classed by the authors among “pleural cancers” and not among “mesotheliomas other than of the peritoneum”, even if they are mesotheliomas arising in an anatomical site different from the peritoneum. I was thus surprised by the observation of 3061 deaths from “mesotheliomas other than of the peritoneum” versus 1205 “classed as pleural cancer” (page 7, 45-55) in 2001-2010. Does this mean that a large number of death certificates (2001-2010) did not specify the mesothelioma site? At page 10: the review by Ngamwong et al has been referenced when discussing the multiplicative joint effect of asbestos and smoking. If I understood correctly their work, Ngamwong et al (2015), having carried out an incomplete review of the literature, concluded that case-control studies showed no deviation from a multiplicative model, but rejected the multiplicative in favour of the additive model based on results from cohort studies. Perhaps the
--

	results of the Synergy pooled analysis of case-control studies (Olsson et al, Epidemiology, 2017) or the review and meta-analysis by El-Zoghbi et al (BMC Public Health, 2017) could be referenced instead. Page 11, 12-14: it is unclear to me which previous estimates the authors were considering when stating “that national burdens of lung cancer from occupational exposure to asbestos may not be so high as previously has been thought”. Reference(s) should be provided. The heterogeneity of results across the 22 job groups selected for this analysis is remarkable, as the authors pointed out. As possible explanations, they mentioned selective exposure to different asbestos varieties and intensity and timing of exposure. In which ways such features of asbestos exposure could explain the findings? For instance, did the jobs with lower ratios entail selective exposure to crocidolite (for which the authors mention a lower ratio, compared with amosite or chrysotile)? Can high PMRs for pleural cancer be considered indices of exposure to amphiboles, or of severe exposures, and, if so, is there evidence of any relationship with the ratio?
--	--

REVIEWER	Fabio Barbone Department of Medicine (DAME), University of Udine, Italy
REVIEW RETURNED	02-Feb-2020

GENERAL COMMENTS	This is an interesting analysis pertaining to a major, world-relevant question: the accuracy of estimating lung cancer cases attributable to asbestos exposure from pleural cancer rates. Although the topic is not original, differently from previously published studies, this manuscript extends data to year 2017 and uses a different technique to adjust for smoking by job title. The major concern I have about this paper relate to the generalizability of its results.  1. Compared to reference [8] this analysis relates to England and Wales and not to Great Britain. However, the authors do not recognize this important spatial difference. Can these results be generalized to Scotland? To Northern Ireland? To Europe? To the World? This issue should be discussed thoroughly and possibly at least the Scottish data should be included. 2. The sub analysis conducted to mesotheliomas (defined in various ways) and related to years 2001-2010 confuse rather than confirm the overall results that should remain focus on pleural cancer. 3. It appears than over this long period of observation (1978-2010) there may be variation of the measure of interest (the ratio between lung cancer attributable to asbestos and number of pleural cancer), my suggestion is that after Figure 2, authors display these job-specific results stratified also by time-period (1988-1990, 1991-2000, 2001-2010). On the other hand the confusing Figure 3 and
---

	the mesotheliomas issue should be omitted or this part of the paper completely reorganized. Based on my comments above, on the actual results of Figure 2 and on the literature, it is of utmost importance to eliminate from the abstract and the discussion the following comment: ‘Together, these two investigations suggest that national burdens of lung cancer from occupational exposure to asbestos may not be so high as previously has been thought.’
--	---

VERSION 1 – AUTHOR RESPONSE

Reviewer: 1

In this paper an original approach has been applied to estimate the lung cancer burden attributable to occupational exposures to asbestos, based on England and Wales mortality data, 1979-2010: (i) occupation-specific estimates have been computed for 22 asbestos-exposed occupational groups, where (ii) the expectation for lung cancer had been indirectly adjusted for group-specific smoking prevalence by accounting for COPD mortality.

I have a few, minor remarks.

In the methods the following definitions have been given: “mesotheliomas other than of the peritoneum (C45.2, C45.7 and C45.9)” (page 6, 54) and “cancer of the pleura (ICD9 163, ICD10 C38.4, C38.8 and C45.0)” (page 6, 12). I understood that pleural mesotheliomas (C45.0) had been classed by the authors among “pleural cancers” and not among “mesotheliomas other than of the peritoneum”, even if they are mesotheliomas arising in an anatomical site different from the peritoneum. I was thus surprised by the observation of 3061 deaths from “mesotheliomas other than of the peritoneum” versus 1205 “classed as pleural cancer” (page 7, 45-55) in 2001-2010. Does this mean that a large number of death certificates (2001-2010) did not specify the mesothelioma site?

RESPONSE:

The reviewer’s understanding is correct. As we highlight in paragraph 6 of the Discussion, before 2001, “... deaths ascribed to mesothelioma without any specified anatomical location (most of which would have been pleural) were classed along with other cancers of unknown origin. Data from 2001-10, when they were assigned to a specific code, indicated that they outnumbered deaths ascribed to pleural cancer more than twofold.”

Pleural mesotheliomas (C45.0) were classified among pleural cancers. In the sensitivity analysis for 2001-2010, we also included deaths from mesothelioma of the pericardium (C45.2), other specified sites (i.e. not pleura, peritoneum or pericardium) (C45.7) and at unspecified site (ICD 45.9). Ideally, we would not have included C45.2 or C45.7, but that was not possible because we had data only for

all mesotheliomas combined (C45), pleural mesothelioma (C45.0) and peritoneal mesothelioma (C45.1). In practice, however, the impact will have been minimal since only a tiny minority of mesotheliomas arise at sites other than the pleura or peritoneum.

At page 10: the review by Ngamwong et al has been referenced when discussing the multiplicative joint effect of asbestos and smoking. If I understood correctly their work, Ngamwong et al (2015), having carried out an incomplete review of the literature, concluded that case-control studies showed no deviation from a multiplicative model, but rejected the multiplicative in favour of the additive model based on results from cohort studies. Perhaps the results of the Synergy pooled analysis of case-control studies (Olsson et al, Epidemiology, 2017) or the review and meta-analysis by El-Zoghbi et al (BMC Public Health, 2017) could be referenced instead.

RESPONSE:

The systematic review by Ngamwong demonstrated “a positive synergistic interaction on an additive scale between asbestos exposure and cigarette smoking in workers developing lung cancer”, but the joint effect of the two exposures fell short of being multiplicative. We now refer simply to the joint effect being more than additive, and have added a second reference in support (both in the Introduction where it is now first mentioned, and in the Discussion).

Page 11, 12-14: it is unclear to me which previous estimates the authors were considering when stating “that national burdens of lung cancer from occupational exposure to asbestos may not be so high as previously has been thought”. Reference(s) should be provided.

RESPONSE:

We have modified the wording to “The lower ratio that we observed suggests that the current burden of lung cancer from occupational exposure to asbestos in Britain may not be so high as previously has been thought”, and have added reference to a recent HSE report that assumed a ratio of one.

The heterogeneity of results across the 22 job groups selected for this analysis is remarkable, as the authors pointed out. As possible explanations, they mentioned selective exposure to different asbestos varieties and intensity and timing of exposure. In which ways such features of asbestos exposure could explain the findings? For instance, did the jobs with lower ratios entail selective exposure to crocidolite (for which the authors mention a lower ratio, compared with amosite or chrysotile)? Can high PMRs for pleural cancer be considered indices of exposure to amphiboles, or of severe exposures, and, if so, is there evidence of any relationship with the ratio?

RESPONSE:

We do not have data on levels of exposure to different types of asbestos at different times in our 22 job groups, and we doubt that such information could be obtained. Therefore, our thinking is necessarily speculative. We do not have evidence that the jobs with the lowest ratios (draughtspersons, chemical engineers, electricians/maintenance fitters) were particularly exposed to crocidolite as compared with, say, dockers, scaffolders or production/maintenance engineers. We have now added differences in smoking habits as another possible reason for the heterogeneity.

Reviewer: 2

This is an interesting analysis pertaining to a major, world-relevant question: the accuracy of estimating lung cancer cases attributable to asbestos exposure from pleural cancer rates.

Although the topic is not original, differently from previously published studies, this manuscript extends data to year 2017 and uses a different technique to adjust for smoking by job title.

RESPONSE:

Please note that we used data to 2010, and not 2017.

The major concern I have about this paper relate to the generalizability of its results.

1. Compared to reference [8] this analysis relates to England and Wales and not to Great Britain. However, the authors do not recognize this important spatial difference. Can these results be generalized to Scotland? To Northern Ireland? To Europe? To the World? This issue should be discussed thoroughly and possibly at least the Scottish data should be included.

RESPONSE:

Analysis was limited to England and Wales rather than the Great Britain because we did not have data for Scotland. However, as we argue in our Discussion, we do not think that the variability in ratios of excess lung cancer to mesothelioma can be explained by limitations of our study methods. If that is accepted, then our conclusion that “The potential for variability in the ratio of excess lung cancer to mesothelioma should be taken into account when estimating population burdens of the disease from occupational exposure to asbestos” will apply to other countries, and not only to England and Wales or the UK..

2. The sub analysis conducted to mesotheliomas (defined in various ways) and related to years 2001-2010 confuse rather than confirm the overall results that should remain focus on pleural cancer.

RESPONSE: As we have noted in our response to Reviewer 1, we think this sensitivity analysis is important because of the high proportion of deaths from mesothelioma that are not coded as pleural cancer where the site of the tumour is not specified on the death certificate. Without it, we think that our findings would be less robust, and we would not have a meaningful estimate of the overall ratio of excess lung cancer to mesothelioma.

3. It appears than over this long period of observation (1978-2010) there may be variation of the measure of interest (the ratio between lung cancer attributable to asbestos and number of pleural cancer), my suggestion is that after Figure 2, authors display these job-specific results stratified also by time-period (1988-1990, 1991-2000, 2001-2010). On the other hand the confusing Figure 3 and the mesotheliomas issue should be omitted or this part of the paper completely reorganized

RESPONSE: When originally preparing our paper, we did consider an analysis stratified into three time periods as the reviewer suggests. However, we decided against it because statistical precision would be reduced by the smaller numbers, and we did not envisage it adding usefully to the conclusions that we could draw. In particular, we were aware that in the sensitivity analysis for 2001-10 (which included all non-peritoneal mesotheliomas), there was again marked variation in the ratios of excess lung cancer to mesothelioma, with the job groups that showed the highest and lowest values much the same as in the main analysis (penultimate paragraph of Results). We accept that the problems with incomplete specification of mesotheliomas on death certificates may be a little confusing for some readers, but that is not a reason to brush them aside. The analysis presented in Figure 3 is needed to justify the main conclusions of the paper.

Based on my comments above, on the actual results of Figure 2 and on the literature, it is of upmost importance to eliminate from the abstract and the discussion the following comment: *'Together, these two investigations suggest that national burdens of lung cancer from occupational exposure to asbestos may not be so high as previously has been thought.'*

RESPONSE:

The reviewer has not highlighted any flaw in our study or evidence in the published literature that calls into question our finding that ratios of excess lung cancer to mesothelioma can vary widely between different occupational groups. It follows that depending on the occupational mix within a population, the burden of lung cancer may not always be as high as has been inferred from calculations that have assumed a relatively high ratio of excess lung cancer to mesothelioma. In particular, we now cite a recent British estimate that assumed a ratio of one. We see no reason to withdraw our conclusion.

VERSION 2 – REVIEW

REVIEWER	Dario Mirabelli Formerly: Cancer Epidemiology, CPO-Piemonte and University of Turin. Now retired I served as expert witness for the public prosecution office in several criminal trials on the death from mesothelioma or lung cancer of asbestos exposed workers
REVIEW RETURNED	29-Feb-2020

GENERAL COMMENTS	All the points I raised have been carefully addressed and I have no additional requests
---

REVIEWER	Fabio Barbone University of Udine, Italy
REVIEW RETURNED	28-Feb-2020

GENERAL COMMENTS	Over this long period of observation (1978-2010) there may be variation of the measure of interest (the ratio between lung cancer attributable to asbestos and number of pleural cancer). Authors should display these job-specific results stratified also by time-period (1988-1990, 1991-2000, 2001-2010). Were the size of specific jobs by decade too small, authors should group jobs in a meaningful way and display the ratio by group and decade at least as an appendix.
--

VERSION 2 – AUTHOR RESPONSE

Reviewer: 1

All the points I raised have been carefully addressed and I have no additional requests

RESPONSE:

Thank you

Reviewer: 2

Over this long period of observation (1978-2010) there may be variation of the measure of interest (the ratio between lung cancer attributable to asbestos and number of pleural cancer). Authors should display these job-specific results stratified also by time-period (1988-1990, 1991-2000, 2001-2010). Were the size of specific jobs by decade too small, authors should group jobs in a meaningful way and display the ratio by group and decade at least as an appendix.

RESPONSE:

As we previously responded, when originally preparing our paper, we did consider an analysis stratified into three time periods as the reviewer suggests. At that time, we decided against it

because statistical precision would be reduced by the smaller numbers, and we did not envisage it adding usefully to the conclusions that we could draw.

However, as the reviewer requests, we have now included Supplementary Table 3 which provides the ratio of excess deaths from lung cancer to observed deaths from cancer of the pleura, by job group and time period. We have added a sentence to the end of the second paragraph of the result section: “For completeness, Supplementary Table 3 shows this job-specific ratio stratified also by time-period (1979-1990, 1991-2000 and 2001-2010).”